# Which Receives More Attention, Online Review Sentiment or Online Review Rating? Spillover Effect Analysis from JD.com

**DOI:** 10.3390/bs14090823

**Published:** 2024-09-15

**Authors:** Siqing Shan, Yangzi Yang, Chenxi Li

**Affiliations:** School of Economics and Management, Beihang University, Beijing 100191, China; shansiqing@buaa.edu.cn (S.S.); chenxili@buaa.edu.cn (C.L.)

**Keywords:** online reviews, spillover effect, interaction effect, consumer behavior, competition, latent Dirichlet allocation

## Abstract

Studies have found that competitive products’ online review ratings (ORRs) have a spillover effect on the focal product’s sales. However, the spillover effect of online review sentiment (ORS) as an essential component of online review analysis has yet to be studied. In this study, we analyze online review content from JD.com using the latent Dirichlet allocation to identify the product attribute topics that consumers are most concerned about. We then construct a baseline regression model of ORS and ORRs to explore the effects of online competitive product reviews on focal product sales. Moreover, we examine how the interaction between ORS and critical factors of online reviews affect sales. Our results indicate that the ORS of competitive products has a negative effect on focal product sales, and the effect is greater than the ORS and ORRs of focal products, respectively. In addition, the ORS of competitive products inhibits the sale of focal products as evaluations of product attributes become more positive or online review usefulness increases. We also find that the effect of ORRs of competitive products is not significant, which may be because clothing, as an experiential product, requires consumers to gain more information about specific usage scenarios before making a decision. This study provides a more accurate basis for consumer decision-making and offers retailers a novel approach to developing marketing strategies.

## 1. Introduction

Consumers often rely on online reviews to inform their purchase decisions, regardless of the platform or product category [1,2,3,4,5]. This strong correlation between online reviews and sales is well documented [6]. However, the purchasing process itself is usually more complex, as consumers tend to compare and evaluate multiple related products before making a final decision. This comparative process is known as “market basket selection”, which is a significant factor influencing sales [7].

The spillover effect is a phenomenon that refers to the impact of events in one environment on those in another seemingly independent environment [8]. When consumers search for products, they tend to examine the online reviews of both the focal product (the one they plan to purchase) and its competitive products simultaneously [6,9]. Recently, cross-product spillovers have attracted considerable attention [10,11,12,13]. Borah et al. found that negative evaluations of one product can extend to another product [14]. Kwark et al. discussed the spillover effect of online review ratings (ORRs) from related products on sales [15]. However, their study on the spillover effect of ORRs overlooked the specific content of online reviews. Therefore, there is still a lack of more detailed research on the spillover effects of online review sentiment (ORS) on product sales.

The value of online reviews (i.e., ORRs and ORS) lies in conveying the voice of consumers through their experiences with the product [16]. However, ORS and ORRs have significantly different characteristics. An ORR provides a quantifiable score that offers a broad assessment, while the ORS consists of detailed textual descriptions that explore specific aspects of the consumer’s experience. Although numerous studies have verified the critical role of ORS in consumers’ purchasing and retailers’ marketing decisions [1,16,17], there is limited research on how ORS affects online competitive products [12]. Moreover, the cognitive theory of multimedia learning holds that multiple-channel communications appear superior to single-channel communications [18,19]. Thus, this study considers the spillover effects of ORS and delves into the effects of the interaction between multiple online review factors and ORS on product sales. We aim to answer the following two questions: (1) Which are consumers more concerned about, ORS or ORR? (2) How does the interaction between ORS and other online review factors of competitive products affect focal product sales?

The significance of investigating these issues from the perspective of spillover effects is apparent. Consumers tend to choose their preferred products from multiple similar options [7]. Consequently, the spillover effects on competitive products can impact consumers’ purchasing decisions. We first select competitive products and focal products based on similarity measures of product characteristics. Next, a sentiment analysis is conducted on a large number of online reviews. Then, we analyze the impact of ORS, ORRs, and other factors from online reviews on the sales of the focal product using a baseline model and interaction model. Our results indicate that the ORS of competitive products has a negative impact on the sales of the focal product, and this impact is greater than that of the ORR of the focal product. However, the effect of the ORRs from competitive products is not significant.

The remainder of the study is organized as follows. First, we review the relevant literature. Second, we build the product set and identify consumers’ relevant product attributes from online reviews using latent Dirichlet allocation. Subsequently, we present the theory and hypothesis. Section 5 describes the methodology of baseline and interaction models. Next, we conduct a practical examination and evaluate the robustness of our model. Finally, we discuss the theoretical and managerial implications and summarize our study’s future prospects.

## 2. Literature Review

Online reviews are increasingly important in understanding users’ evaluation behavior, providing both users and organizations with improved opportunities for decision-making. On the one hand, Dimoka et al. have demonstrated that online product reviews impact consumer behavior because they help reduce uncertainty about products [20]. On the other hand, understanding how the information embedded in online reviews drives sales can help enterprises predict consumer behavior, promote new products, and retain shoppers [21,22]. Recently, several studies have examined the relationship between online reviews and product sales. For example, Forman and Ghose demonstrated that positive online reviews that include reviewer profile details have a beneficial effect on product sales [23]. Gutt et al. reviewed the effects of online reviews on economic results, such as sales [24]. Zhu and Zhang discussed how product and consumer characteristics regulate the impact of online reviews on product sales [25]. However, online reviews come in various forms in real-world scenarios. Online review ratings [26], review texts [16], reviewer profiles [23], and product characteristics [25] have been shown to impact product sales.

As a typical factor in review text, the online review sentiment of a product is considered an influential indicator of product sales [27]. Studies suggest that the more positive the online reviews about the product, the higher the sales [1]. For example, Li et al. discovered that consumers are more inclined to purchase products when experiencing positive or negative emotions compared to uncertain emotions [21]. Zhang and Qiu found that positive online reviews have a more significant impact on sales than negative online reviews [28].

The spillover effect refers to the accidental impact of an event in one environment on other individuals in a different, indirectly related environment [29]. Spillover effects are widespread in marketing and are increasingly attracting the attention of scholars [14,30]. For example, Lewis and Nguyen found that competitors’ advertisements have a strong positive spillover effect on similar products [11]. Liang et al. empirically studied the spillover effect of recommendations in the mobile app market [31]. Further, Wu et al. researched the brand spillover effects considering company and market characteristics [32]. Joe and Oh studied the indirect spillover effects among companies within the same group [33].

As for online reviews, recent studies have shown that online reviews can spread to competitors and adversely affect their sales [34], with most of the research focusing on the spillover effect of online review ratings [35,36]. For example, Deng et al. created a two-factor fixed-effects model to examine the spillover effect of online review ratings [35]. Kwark and Chen analyzed the spillover effect of relevant online reviews on the sales of focal products [36] and found that the online review ratings of alternative products have a negative spillover effect. However, there are few studies have considered the spillover effect of sentiments in online reviews. In addition, it remains inconclusive whether the spillover of review ratings or sentiment from competing products has a greater impact on product sales.

## 3. Theory and Hypotheses

According to the cognitive–emotional theory, both cognitive (such as attribute) and affective aspects (such as sentiment) influence consumer decision-making [37]. There is a consensus among scholars that both cognition and emotional cognition should be considered to study their joint effects on consumer decision-making [38]. In this section, the role of the ORS of competitive products on the focal product purchase is discussed first (Hypotheses 1a and 1b). Then, we hypothesize the role of competitive products’ cognition aspects (product attributes) and ORS in the purchasing of a focal product (Hypotheses 2). In addition, according to the uncertainty reduction theory, if consumers lack knowledge about a product, they will proactively seek other information to diminish the uncertainties [39]. Detailed and highly recommended online reviews will lessen the uncertainty of shopping, thereby impacting consumers’ purchases. Thus, we also hypothesize the roles of the ORPs (Hypotheses 3) and ORU (Hypotheses 4) of competitive products in purchasing a focal product.

### 3.1. Online Review Sentiment

Online reviews are a valuable source of information for consumers when making purchasing decisions [40]. Prior studies have regarded online reviews as signals of product quality [2,5]. By reading online reviews, consumers consider other competitive products to find the highest utility [15]. Studies have found negative relationships among demand changes in competitive products. The marketing activity of one product, such as promotion, may have a negative effect on other competitive products [41]. As part of online review feedback, online review ratings represent a comprehensive scoring for online products, demonstrating the impact on product demand. Kwark et al. explored the effect of the online review ratings of competitive products on the sales of focal products [15]. However, ORRs do not capture the product’s specific characteristics.

Consumers can generate and trigger specific emotional responses during the consumption experience or product usage [42]. Hence, online reviews exhibit a clear emotional tone, uncovering consumers’ positive and negative perspectives on the product’s characteristics [43]. When potential consumers receive a large amount of positive information about a product, it enhances their subjective perception of the product’s value and assists potential consumers in making decisions swiftly [44,45]. In contrast, when consumers read many negative online reviews, they may perceive the product with many shortcomings, dampening their desire to purchase [40]. Thus, when the relative attractiveness of the focal product is weakened, consumer preferences shift to competitive products [2]. Considering the positive relationship between the ORS (ORRs) and product demand, as well as the negative relationship with competitive products in consumer purchase decisions, we propose the following hypotheses:

**H1a.** *The ORS of competitive products has a negative effect on the sales of the focal product*.

**H1b.** *The ORS of competitive products has a more negative impact on the sales of focal products than ORRs*.

### 3.2. Role of Online Review Characteristics

#### 3.2.1. Product Attributes

High-quality online reviews can provide potential consumers with information on product attribute (PA) performance, thereby enhancing the potential consumers’ cognition of the product [46,47]. Many online reviews focused on product quality suggest that people are interested in or generally interested in the critical attributes of products [7]. Previous studies have investigated how product attributes affect consumer purchases. Wells et al. investigated the relationship between purchase intention and perceived product quality in the online environment, finding that the relationship is consistently significant across a series of cases [48]. Consumers can identify product quality through functional online reviews to reduce purchase risk and product uncertainty [34]. Jang et al. added two more factors and examined the attribute values of online product reviews that can lead to further sales [49].

Product design and performance affect consumers’ cognition and emotions. Sun et al. highlighted the significant association between product attributes and consumer emotions [43]. Online reviews with a positive sentiment tendency toward product attributes facilitate product diffusion [50]. If the online review contains a positive opinion, it lessens consumer worries about the product and increases cognition of its benefits [42]. However, low-quality online reviews with negative sentiment tendencies lower consumers’ perceived product quality and thus inhibit product diffusion [43]. Thus, we propose the following hypothesis:

**H2.** *The interaction between the ORS and PA of competitive products has a negative effect on the sales of the focal product*.

#### 3.2.2. Online Review Photos

Online review photos (ORPs) are posted by consumers when commenting on online products. Compared with online review text, ORPs contain richer and more vivid information and are easily manageable [51]. Given the intangible nature of product experience, scholars have just begun to realize the importance of understanding and managing consumers’ visual attention as a pivotal tool for sharing experiences [52]. Visual cues may arouse greater interest in relevant online reviews and stimulate uninterested consumers to read more carefully. Ma et al. found that combining online review text and ORPs can improve the acceptance of online reviews [53]. According to the multimedia cognitive theory, multi-channel communication seems superior to single-channel communication when related signals are added between channels [19]. Therefore, taking into account the visual content in online reviews can improve consumer cognition and learning abilities [54].

Another stream of research has revealed that a more significant number of ORPs can expand the quantity and diversity of pictorial cues, which evokes stronger emotions corresponding to the ORPs [55]. As a supplementary visual representation to online review text, the display of photo details can intensify the existing emotional tendency of the online review text. If an online review text containing photos is positive, it reduces consumer concerns about the product while increasing awareness of its benefits [53]. Accordingly, the following hypothesis is posited:

**H3.** *The interaction between the ORS and ORPs of competitive products has a negative effect on the sales of the focal product*.

#### 3.2.3. Online Review Usefulness

Online reviews are helpful when customers gain clear information about the quality and performance of a product [56]. Online review usefulness (ORU) is commonly used for gauging a review’s utility [17,57]. Users are increasingly suspicious of the authenticity of online reviews, given the possibility of manipulation [58]. Existing studies have presented views and suggestions regarding the quantitative analysis of the usefulness of online consumer reviews. Some studies have adopted the presence of several helpful votes as the primary measure of ORU [59,60]. Cui et al. assessed product popularity based on the review votes of consumers with product experience [61]. Hong et al. suggested that the acceptance of a review is subject to the number of votes it obtains [62].

Consumers are more receptive to and influenced by online reviews perceived as more helpful. Chen et al. found that more helpful and critical online reviews have a greater impact on sales than other online reviews [63]. If a positive online review is considered useful, the content is trusted by potential customers, thus raising awareness of the product’s benefits [64]. Conversely, useful online reviews with negative emotional tendencies decrease consumers’ desire to purchase. Therefore, we propose the following hypothesis:

**H4.** *The interaction between the ORS and ORU of competitive products has a negative effect on the sales of the focal product*.

## 4. Methodology

### 4.1. Product Selection

We chose online clothing reviews on JD.com as the object of study. JD.com is one of China’s largest B2C online retailers, holding a leading market share [43]. Moreover, clothing consumption is a significant factor in expanding national consumption and an essential embodiment of quality upgrading. According to the Statista statistics 2022, U.S. clothing and clothing accessories sales amounted to approximately 312.4 billion U.S. dollars. Therefore, we select clothing from JD.com as the main online shopping website for data collection.

When consumers purchase a product, they consider online reviews of several similar (competitive) products. Focal products are items that retailers aim to sell through in-store marketing [15]. Competitive products are often consumed along with focal products, and the demand for competitive products is usually negatively correlated with that for focal products [43]. The spillover effects appear stronger when competitive and focal products are displayed together [15]. Several studies have provided different definitions for identifying competitive products [2,41]. In this study, we survey the Credano.com platform to explore consumer preferences when purchasing products online. After removing responses with missing data, we obtain 188 valid questionnaires. Participants are asked about their methods for browsing competitive products while shopping. Preliminary test outcomes show that consumers generally search for related competitive products using direct product searches (35.5%), viewing product rankings (39.4%), opting for platform-recommended products (23.7%), and through suggestions from blogs, friends, or through social media channels (1.4%). Thus, according to JD.com, there are three sources from which we obtain the competitive product set: product searches, sales ranking lists, and platform-recommended products. After obtaining the product set, we measure the individual attribute similarity between the products for clustering. Thus, in each category, when a product is a focal product, the remaining products are divided into competitive products for the focal product.

### 4.2. Data Collection

There are two datasets we collect—online product reviews and product sales. We use the daily sales ranking of products provided by the third-party platform Boshi.com. Figure 1 shows an example of an online review of clothing products on JD.com. Our database contains nearly 250,300 online reviews from 1 November 2022 to 1 May 2023. We require the number of daily online reviews in the analyzed period to be substantial and of high quality for every product. Thus, data screening and cleaning based on rule definition are conducted, given the unstructured textual format of the original data. Consumers usually distrust products with few online reviews because it is difficult to search for useful information from only a few online reviews [65]. Therefore, we delete products that have not been reviewed or have few online reviews. Finally, we identify 17 different types of clothing containing 269 online products, with a total of 46,116 valid samples.

### 4.3. Product Attribute Analysis

Latent Dirichlet allocation is one of the most popular topic models in the machine learning field [66]. It utilizes a probabilistic structure to extract topics from extensive review content [66]. In this work, we use latent Dirichlet allocation to identify product attribute topics of interest to consumers and employ the Gibbs sampling technique to estimate parameters [7]. We add the perplexity/coherence with respect to the number of topics in Figure 2. From the results, we can find that the best topic effect occurs when the number of topics is five. Meanwhile, Table 1 presents the details of the five topics. We can observe that the top 10 words generally differ across the five topics identified by LDA, corresponding to distinct themes: quality, size, fabric, design, and comfort. However, there are indeed some words that contribute to multiple topics, i.e., word “fit” for the size and design themes, and the word “breathable” for the fabric and comfort themes.

### 4.4. Variable Measurements

We utilize the SnowNLP package to determine the sentiment score of each online review, ranging from 0 (negative) to 1 (positive). The larger the value, the greater the intensity of positive sentiments. Online review ratings, on the other hand, represent users’ overall assessment of the product, ranging from 1 to 5.

As a visual expression of auxiliary online review text, ORPs are quantified by the number of photos in each online review. Considering that each product will receive online reviews from multiple reviewers, online review usefulness (ORU) is measured by the number of likes each online review receives. For PA, we use the aforementioned LDA method to obtain the attribute words under each topic and determine the sentiment tendency of each topic based on a sentiment dictionary. Then, we count the number of positive and negative product attributes (quality, size, fabric, design, and comfort) mentioned in each online review. Therefore, the calculation formula for PA is as follows:(1)$${PA\_comp}_{i}=\frac{1}{N}\sum_{j=1}^{N}\left({Att\_pos}_{ij}-{Att\_neg}_{ij}\right)$$
where ${Att\_pos}_{ij}$ is the number of positive topics related to feedback on product attributes in the online reviews of competitive product *j* for each focal product *i*. ${Att\_neg}_{ij}$ represents the number of negative topics related to feedback on product attributes in the online reviews of competitive product *j* for each focal product *i*. To explore the impact of the spillover effect of online competitive products, we use the clothing sales ranking from the JD.com platform as the dependent variable. The independent variables include the online review characteristics (ORS, ORRs, ORPs, PA, and ORU) and control variables (volume, promotion, and brand). Table 2 and Table 3 describe the primary variables and corresponding descriptive statistics.

### 4.5. Empirical Specification

Here, we summarize our empirical model. First, we focus on the effects of the ORS and ORRs of competitive products on focal product sales in the baseline model (Hypotheses 1a and 1b). Second, we investigate the interaction effect of PA and ORS (Hypothesis 2). Third, we examine the interaction effect of ORPs and ORS (Hypothesis 3). Finally, we study the interaction effect of ORU and ORS (Hypothesis 4).

#### 4.5.1. Baseline Model

We establish the baseline model to study the effects of the ORS and ORRs of competitive products on the sales of focal products. Since the online review information that consumers view has already been published, we introduce a lag period for the explanatory variables [43]. In similar competitive product category sets, when a product is identified as a focal product, the rest are classified as competitive products relative to the focal product. Thus, the calculation formula is as follows:(2)$${sales}_{it}=\lambda_{t}+\beta_{1}{sentiment\_focal}_{it-1}+\beta_{2}{sentiment\_comp}_{it-1\;}\;\;\;\;\;\;\;\;\;{+\beta }_{3}{rating\_focal}_{it-1}+\beta_{4}{rating\_comp}_{it-1}+\beta_{5}{control}_{it}+a_{i}+\varepsilon_{it}$$
where ${sales}_{it}$ represents the sales ranking of focal products *i* in week *t*. The variable ${sentiment\_focal}_{it-1}$ indicates the mean sentiment score of the focal products’ online reviews with a lag of one. The variable ${sentiment\_comp}_{it-1}$ represents the mean sentiment score of competitive products’ online reviews in week *t* − 1. The variable ${rating\_focal}_{it-1}$ is the mean number of online review ratings of focal products in week *t* − 1, and ${rating\_comp}_{it-1}$ is the mean number of online review ratings of competitive products in week *t* − 1. ${control}_{it}$ is the control variable set, including ${volume}_{it},\;{\;promote}_{it},\;\mathrm{and}\;{brand}_{it}$. Specifically, ${volume}_{it}$ is the mean volume of online reviews of focal products in week *t*. ${promote}_{it}$ is a dummy variable; if the focal product *i* is on promotion, ${promote}_{it}$ = 1; otherwise, it is 0. ${brand}_{it}$ is the mean number of product brand votes on chinapp.com, which serves as a platform resource for consumers or operators to inquire and learn about brand data. Moreover, $\lambda_{t}$ is the intercept term, $a_{i}$ denotes unobserved factors, and we used robust t-statistics to treat unknown error terms $\varepsilon_{it}$.

#### 4.5.2. Interaction Model

The interaction effect among online review characteristics also impacts product sales [58]. Therefore, considering online review sentiment as an example, we build the interaction effect model rooted in the baseline model:(3)$$\begin{array}{cc} {sales}_{it}= & \lambda_{t}+\beta_{1}{sentiment\_focal}_{it-1}+\beta_{2}{sentiment\_comp}_{it-1}\\ & +\beta_{3}{rating\_focal}_{it-1}+\beta_{4}{rating\_comp}_{it-1}+\beta_{5}{control}_{it}\\ & \;\;\;+\beta_{6}{factor}_{it-1}+{\beta_{7}sentiment\_comp}_{it-1}\times {factor}_{it-1}+a_{i}+\varepsilon_{it} \end{array}$$
where ${factor}_{it-1}=\{{ORP\_comp}_{it-1},{PA\_comp}_{it-1},{ORU\_comp}_{it-1}\}$, and ${ORP\_comp}_{it-1}$ is the mean number of photos from competitive products’ online reviews in week *t* − 1. ${PA\_comp}_{it-1}$ is the mean product polarity attribute difference from competitive products’ online reviews in week *t* − 1. And ${ORU\_comp}_{it-1}$ is the mean number of useful votes from competitive products’ online reviews in week *t* − 1. ${sentiment\_comp}_{it-1}\times {factor}_{it-1}$ captures the interaction effects of ORS and ${ORP\_comp}_{it-1}$, ${PA\_comp}_{it-1}$, and ${ORU\_comp}_{it-1}$ respectively.

## 5. Results

In this section, to provide a deep explanation of how the ORS of competitive products impacts focal products’ sales, we explore the direct and interaction effects using empirical models. Moreover, we conduct a robustness test in Section 5.2. In particular, we perform an extended analysis by selecting a second type of product to further validate the findings of our study.

### 5.1. Regression Results

Our fixed-effects estimation results of the baseline model are presented in Table 4. Model 1 introduces the ORS and ORRs for both focal and competitive products. Results show that the coefficients of sentiment_focal and rating_focal are significantly positive, whereas the coefficient of sentiment_comp is significantly negative, and that of rating_comp is not significant. The findings suggest that both the ORS and ORRs of the focal product have a positive impact on its sales, while the ORS of competitive products negatively impacts the sales of the focal product. According to uncertainty reduction theory, this may be because consumers seek more information to increase their awareness of products during the purchase process, leading them to pay more attention to the online review content (sentiment) of competitive products than to ratings. In addition, the negative effect of sentiment (−0.1138, *p* < 0.01) for competitive products on the sales of a focal product is stronger than the effect of the focal product itself (0.1011, *p* < 0.01). Thus, Hypothesis 1a is supported, but Hypothesis 1b is not. Noticing that the coefficients appear modest, we also consider their practical significance. While a unit change in the dependent variable may lead to a 0.1 shift in ranking, in practice, given JD.com’s large sales volume, even a slight change in ranking could result in a significant impact on sales. This observation aligns with the findings in Kwark’s research [15].

Next, in Models 2 to 4, we incorporate the ORPs, PA, and ORU of competitive products separately. The results reveal that the ORPs, PA, and ORU of competitive products have a significant negative impact on the sales of focal products. Among them, the negative effect of PA (−0.0283, *p* < 0.05) is relatively weak, while that of ORU (−0.0814, *p* < 0.01) is stronger. Model 5 presents the results for all the variables. We find that the primary variables significantly affect the sales of focal products, with the exception of rating_comp. Then, we take Model 5 as an example to analyze the results of control variables. volume, promotion, and brand all have a positive impact on product sales. First, a large number of online reviews can reduce consumer uncertainty and promote product purchases. Second, promotion serves as a value incentive signal and can drive sales growth in the short term [43]. Additionally, brand trust plays a key role in consumer purchasing decisions, with a strong brand effect typically leading to higher sales and market share [14].

We proceed to examine the interaction effects of online review metrics under ORS. Table 5 presents the regression results of the interaction models. In Model 6, the interaction term of the ORS and ORPs (sentiment_comp × ORP_comp) of competitive products has no significant impact on the focal product’s sales, not supporting Hypothesis 2. As expected, results in Model 7 illustrate that the interaction term of the ORS and PA (sentiment_comp × PA_comp) of competitive products significantly negatively affects the focal product’s sales (−0.0413, *p* < 0.05). Thus, Hypothesis 3 is supported, revealing that a greater PA value suggests the high-quality performance of the product, thereby promoting consumer purchases. The results reported in Model 8 indicate that the interaction term of the ORS and ORU (sentiment_comp × ORU_comp) has a significant impact on the focal product’s sales (−0.0993, *p* < 0.01), supporting Hypothesis 4. Model 9 incorporates the analysis of all interaction terms. The results illustrate that all primary and interaction variables are consistent with those from the stepwise regression. In addition, the interaction effects of ORU (−0.0884, *p* < 0.01) have a greater effect than those of PA (−0.0307, *p* < 0.1).

**Table 5 behavsci-14-00823-t005:** The results of interaction effects.

Variables	Model 6	Model 7	Model 8	Model 9
sentiment_focal	0.0963 *** (7.08)	0.0964 *** (7.11)	0.091 *** (6.68)	0.0916 *** (6.72)
sentiment_comp	−0.1049 *** (−4.94)	−0.1195 *** (−5.6)	−0.1425 *** (−6.1)	−0.1493 *** (−6.16)
rating_focal	0.0266 ** (2.43)	0.0272 ** (2.49)	0.0248 ** (2.27)	0.0254 ** (2.33)
rating_comp	−0.0156 (−0.95)	−0.0153 (−0.94)	−0.0114 (−0.7)	−0.0116 (−0.71)
ORP_comp	0.0301 (1.5)	0.0305 * (1.75)	0.0247 (1.41)	0.0255 (1.28)
PA_comp	0.0253 * (1.76)	0.0411 *** (2.6)	0.0292 ** (2.03)	0.0405 *** (2.57)
ORU_comp	0.0738 *** (4.15)	0.0733 *** (4.19)	0.1006 *** (5.23)	0.0972 *** (5)
sentiment_comp×ORP_comp	−0.0097 (−0.47)			−0.0066 (−0.32)
sentiment_comp×PA_comp		−0.0413 ** (−2.4)		−0.0307 * (−1.75)
sentiment_comp×ORU_comp			−0.0993 *** (−3.32)	−0.0884 *** (−2.87)
volume	0.0262 * (1.66)	0.026 * (1.66)	0.0282 * (1.8)	0.0279 * (1.77)
promotion	0.0662 *** (5.4)	0.0689 *** (5.62)	0.0636 *** (5.22)	0.0659 *** (5.46)
brand	0.0261 * (1.76)	0.0241 (1.64)	0.0192 (1.3)	0.0185 (1.25)
#products	269	269	269	269
#online review	250,300	250,300	250,300	250,300
N	46,116	46,116	46,116	46,116
R-squared	0.1502	0.1551	0.163	0.1654

Note. Robust t-statistics in parentheses. * *p* < 0.1; ** *p* < 0.05; *** *p* < 0.01. #products: the number of products. #online review: the number of online reviews.

### 5.2. Robustness Checks

Next, we carry out additional analyses to evaluate our model’s robustness, focusing on the removal of some samples, random selecting, and extension analysis. Here, we detail the robustness checks performed.

***Remove some samples.*** Considering that the data includes the e-commerce platform launch of the shopping festival (11 November and 12 December), where large-scale promotions are carried out through the distribution of coupons or combination discounts, we remove the extreme data affected by these shopping festivals for a robustness check. The results in Model I show that the significance of the primary independent variables stays consistent with the previous results after removing the shopping festival samples, indicating the robustness of the regression model.

***Random selecting.*** There are usually several competitive products for each focal product. Thus, we change the calculation method for the variable sentiment_comp by randomly selecting products to be averaged from the competitive product set of the focal product instead of taking the average of all competitive products as before. The results in Model II show that the significance of the independent variables stays consistent with the previous results after changing sentiment_comp, indicating the robustness of the regression model.

***Extension analysis.*** We further select “mobile phones” on the JD platform as a different product type from “clothing” to explore the validity of our research results. This distinction is based on the classification of products into search and experience categories. Mobile phones align with the characteristics of search products, while clothing trends more toward experience products. As shown in the fourth column of Table 6, the impact of mobile phones is comparatively greater than that of clothing. For example, the coefficient of sentiment_comp increases from 0.1178 (clothing) to 0.1249 (mobile phones), with the significance level remaining unchanged. Consequently, these results further validate the robustness of the study’s findings.

## 6. Discussion and Implications

### 6.1. Discussion

This paper explores the impact of the online reviews of competitive products on the sales of focal products and further analyzes the distinct roles of ORS and ORRs in influencing product sales. In addition, we examine the moderating role of three online review characteristics (ORPs, PA, and ORU). The study produces several key findings:

First, the results indicate that the ORS of competitive products has a negative spillover effect on the sales of the focal product, and this impact surpasses that of the focal product’s own ORS. This finding is consistent with Kwark’s research [15], which demonstrates that, by combining demand effects and cross-product evaluations, online reviews in a competitive environment can generate spillover effects. In our study, as the ORS of competitive products becomes more positive, potential consumers’ intentions to purchase these competitive products increase (emotion effect), while their intentions to purchase the focal product decline (demand effect). Additionally, behavioral economics suggests that people often focus on differences from reference points and make relative judgments about relevant aspects. Consequently, in our study, online reviews of competitive products can act as context for assessing the focal product [67]. For example, suppose the ORS of the focus product is 0.6; when the ORS of a competitive product is higher than 0.6, the purchase intention for that competitive product tends to increase.

Second, we identify a series of key factors that moderate the impact of ORS on purchasing behavior. Zhai’s research indicates that the content characteristics of online reviews positively impact product sales [37]. In our study, we find that the individual variables ORPs, PA, and ORU have distinct impacts on the sales of the focal product within the interaction model, i.e., PA and ORU have a certain moderating effect, while ORPs do not. PA and ORU can intuitively help consumers quickly judge the quality of online reviews [43,68,69]. For PA, when consumers receive clear information about product quality and performance, they are more likely to develop a favorable impression and be prompted to purchase. For ORU, since it is typically measured by the number or proportion of helpful votes [62], a high ORU of positive online reviews indicates that these reviews are perceived as authentic and persuasive. However, the reason why ORPs do not have a moderating effect is that, in practical situations, there are a large number of irrelevant, repetitive, or blurry photos in online reviews, which reduces consumers’ trust in the reviews [65]. This observation aligns with Che et al.’s suggestion that high levels of skepticism in developing and emerging countries can lead to lower purchase willingness [70].

Finally, we perform additional analyses in the robustness checks section by selecting “mobile phones” on the JD.com platform, a product type distinct from “clothing”, to further explore the validity of our findings. The experimental results in Table 6 show that online reviews for both types of products generate spillover effects in a competitive environment. We also find that, compared to clothing, the spillover effect of online reviews is relatively larger on mobile phones. Generally, consumers have higher demands and stronger motivations regarding the quality of high-priced products [43]. As a typical search product, mobile phones rely more heavily on online reviews to reduce uncertainty.

### 6.2. Theoretical Implications

Our research makes several theoretical contributions. First, we provide new evidence highlighting the importance of considering spillover effects in evaluating the impact of online reviews on competitive products. Although existing literature has explored the spillover effects across products [71,72,73], most studies have focused on the impact of factors such as brands, promotions, and advertisements. There has been limited exploration into how spillover effects of online reviews influence product sales, and even fewer studies have investigated the impact of the content characteristics of these reviews. Therefore, from the perspective of spillover effects, we analyze the ORS and ORRs of competitive products, enriching the related literature and research methodology.

Second, the literature has examined the impact of spillover effects in the context of ORRs [74,75]. By investigating the impact of the ORS (online review content) and ORRs of competitive products on focal product sales, we extend previous research on the spillover effects of online review content. The results indicate that the ORS of competitive products receives more attention than the ORRs. This provides important theoretical support for a deeper understanding of the impact mechanism of online review content spillover effects on product sales.

Third, Floyd et al. have argued that the interactions between online review metrics have a more intricate impact on sales than the direct effects themselves [76]. In this paper, we evaluate the moderating role of online review content characteristics in the spillover effect. Specifically, on the basis of ORPs and ORU, we introduce another product attribute factor (PA). The results show that, in the context of competitive products, the interaction between PA and ORS negatively affects the sales of the focal product, which further extends Jang et al.’s study from the spillover effect perspective [77]. Overall, our research provides a detailed analysis of how the content characteristics and sentiment interactions in online customer reviews affect product sales, thus filling a gap in the literature.

### 6.3. Managerial Implications

Apart from theory, this research also offers several practical contributions. First, the online review content (e.g., ORS) of competitive product receives more attention than the ORRs. Therefore, for sellers, it is essential to prioritize the emotional expressions found in online review content to promptly improve product attributes and optimize after-sales services. These improvements can enhance user experience and ultimately promote customer purchasing behavior.

Second, research indicates that the spillover of the sentiment of competitive online reviews significantly impacts focal product sales. Therefore, sellers can optimize online review management systems by developing appropriate promotional strategies. For example, they can prioritize the display of positive online review content, provide a standardized review template, improve interactions with consumers, and quickly respond to negative online reviews.

Finally, the quality of online reviews (ORPs, PA, and ORU) positively impacts product sales. According to the research of Xie et al., online reviews with clear sources are more likely to be trusted [78]. Therefore, moderately disclosing the shopping information of reviewers (such as identity, shopping time, after-sales situation, etc.) can help improve the credibility of online reviews. In addition, the platform also needs to monitor user online reviews strictly and ban accounts that frequently experience shopping anomalies.

### 6.4. Limitations and Future Work

The shortcomings of our study and future research directions are as follows: First, from the perspective of the research objective, we have selected one online platform with certain limitations. The impact of online reviews may vary across different product types and platforms. Future research will consider more products and cross-platform experiments to provide enterprises with a broader range of online review marketing recommendations. Second, factors such as the number, variance, and reviewers of online reviews have yet to be discussed in detail and will be incorporated into the experiment to establish a more perfect framework. Third, we have not differentiated or addressed the duration of the spillover effects of ORS. Moving forward, the long-term and short-term spillover effects will be included in the framework for further analysis and discussion.

## 7. Conclusions

This paper investigates how the ORS and ORRs of competitive products affect the sales of focal products. Using the LDA, we extract five product attribute topics from online clothing reviews: quality, size, fabric, design, and comfort. Then, we conduct regression analyses on both baseline and interaction models. Our study indicates that the ORS of competitive products has a negative impact on the focal product’s sales, and the effect is greater than that of the focal product’s ORS. However, the ORR effect of competitive products is not significant. This may be because clothing is an experiential product, and consumers need to know more about the specific usage scenarios of the product. Moreover, we underscore the complexity of the interaction effects of online reviews. These findings suggest that the ORS of the competitive products inhibits the sales of the focal products with an increase in PA or ORU.

## Figures and Tables

**Figure 1 behavsci-14-00823-f001:**
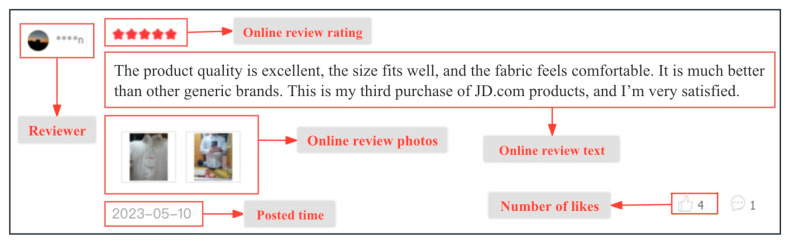
An example of an online review on JD.com.

**Figure 2 behavsci-14-00823-f002:**
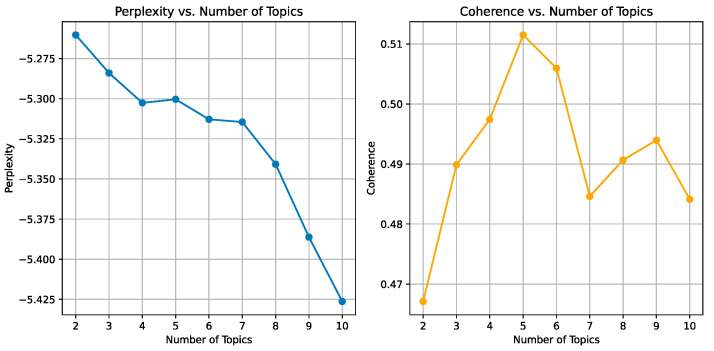
Perplexity/coherence with respect to the number of topics.

**Table 1 behavsci-14-00823-t001:** Results of the topic classification (taking shirts as an example).

Topic	Keywords	Online Review
Quality	Quality (0.16), craftsmanship (0.13), good (0.095), texture (0.090), color discrepancy (0.076), fading (0.054), acceptable (0.052), threads (0.043), detail (0.036), washing (0.036)	The quality is pretty good, the collar stays crisp. After wearing it all summer, it is still good as new.
Size	Fit (0.088), right size (0.062), size (0.060), well-fitting (0.058), standard (0.052), size appropriate (0.052), off-size (0.045), correct size (0.039), accurate (0.038), suitable (0.035)	The clothes have arrived quickly, are the right size, and are well made. It is a satisfying shopping experience.
Fabric	Fabric (0.084), texture (0.075), wrinkles (0.069), pure cotton (0.061), material (0.054), wrinkle-resistant (0.054), breathable (0.050), delicate (0.034), materials used (0.020), skin-friendly (0.020)	The material feels comfortable, even better than some big brands, and the price is affordable. I will definitely come again to purchase.
Design	Cut (0.12), like (0.11), color (0.095), fit (0.061), fashionable (0.055), versatile (0.046), good-looking (0.037), slim-fitting (0.037), style (0.032), on-body effect (0.031)	Fast delivery speed, fashionable clothes design with no redundant elements, very slim.
Comfort	Comfortable (0.13), comfy (0.11), soft (0.090), breathable (0.034), lightweight (0.027), supple (0.027), comfortable to wear (0.026), breathability (0.025), well-fitting (0.024), easy to wear (0.021)	I bought the size 39. 100% long-staple cotton body wear is more comfortable. However, the color choice is a bit less.

**Table 2 behavsci-14-00823-t002:** Primary variables.

Variables	Descriptions
Dependent variable	
sales	Sales ranking of focal product (range: 0+)
Independent variables	
sentiment_focal	Mean sentiment score of focal product’s online reviews (range: 0–1)
sentiment_comp	Mean sentiment score of competitive products’ online reviews (range: 0–1)
rating_focal	Mean rating of focal product (range: 0–5)
rating_comp	Mean rating of competitive products (range: 0–5)
ORP_comp	Mean number of photos from competitive products’ online reviews (range: 0+)
PA_comp	Mean product polarity attribute difference from competitive products’ online reviews (range: 0–5)
ORU_comp	Mean number of useful votes from competitive products’ online reviews (range: 0+)
Control variables	
volume	Mean volume of focal product’s online reviews (range: 0+)
promotion	Whether a product is under promotion (unit: 0, 1)
brand	Mean number of brand votes of focal product (range: 0+)

**Table 3 behavsci-14-00823-t003:** Descriptive statistics.

Variables	Minimum	Maximum	Mean	Std. Dev.
sales	2	193	10.004	13.649
sentiment_focal	0	0.999	0.927	0.111
sentiment_comp	0	0.999	0.895	0.076
rating_focal	1	5	4.95	0.397
rating_comp	1	5	4.94	0.116
ORP_comp	0	4.376	0.589	0.754
PA_comp	−1	5	3.819	1.477
ORU_comp	0	33.21	1.231	1.934
volume	7	243	38.36	2.895
promotion	0	1	0.953	0.211
brand	282	10,960	3773.67	2924.81

**Table 4 behavsci-14-00823-t004:** The results of spillover effects.

Variables	Model 1	Model 2	Model 3	Model 4	Model 5
sentiment_focal	0.1011 *** (7.39)	0.1032 *** (7.56)	0.1002 *** (7.34)	0.0947 *** (7.02)	0.0963 *** (7.08)
sentiment_comp	−0.1138 *** (−5.82)	−0.1198 *** (−4.75)	−0.1181 *** (−6.01)	−0.1123 *** (−5.81)	−0.1043 *** (−5.1)
rating_focal	0.0267 ** (2.41)	0.0269 ** (2.43)	0.0262 ** (2.36)	0.0271 ** (2.46)	0.0266 ** (2.43)
rating_comp	−0.0103 (−0.63)	−0.0075 (−0.45)	−0.0141 (−0.85)	−0.0148 (−0.91)	−0.0156 (−0.95)
ORP_comp		−0.0419 ** (2.41)			−0.0312 * (1.78)
PA_comp			−0.0283 ** (1.96)		−0.0253 * (1.76)
ORU_comp				−0.0814 *** (4.73)	−0.0735 *** (4.19)
volume	0.0484 *** (3.18)	0.0404 ** (2.6)	0.0431 *** (2.79)	0.0355 ** (2.32)	0.0261 * (1.66)
promotion	0.0818 *** (6.84)	0.0789 *** (6.57)	0.0813 *** (6.8)	0.0674 *** (5.51)	0.0661 *** (5.41)
brand	0.0394 *** (2.71)	0.0385 *** (2.65)	0.0404 *** (2.78)	0.0241 * (1.83)	0.0259 * (1.76)
#products	269	269	269	269	269
#online review	250,300	250,300	250,300	250,300	250,300
N	46,116	46,116	46,116	46,116	46,116
R-squared	0.1114	0.123	0.1163	0.1405	0.1501

Note. Robust t-statistics in parentheses. * *p* < 0.1; ** *p* < 0.05; *** *p* < 0.01. #products: the number of products. #online review: the number of online reviews.

**Table 6 behavsci-14-00823-t006:** Robustness checks.

Variables	Remove Some Samples	Random Selecting	Mobile Phone
sentiment_focal	0.1003 *** (7.24)	0.0891 *** (5.76)	0.1136 *** (2.71)
sentiment_comp	−0.1178 *** (−5.96)	−0.1218 *** (−5.31)	−0.1249 *** (−2.63)
rating_focal	0.0328 *** (2.91)	0.0362 *** (2.65)	0.0834 ** (2.03)
rating_comp	−0.0101 (−0.59)	−0.0391 (1.62)	−0.0957 * (−1.89)
volume	0.0306 * (1.92)	0.023 (1.36)	0.1084 ** (2.45)
promotion	0.1022 *** (6.95)	0.1036 *** (6.02)	0.2658 ** (2.22)
brand	0.0331 ** (2.28)	0.0556 *** (3.22)	3.1471 * (1.77)
R-squared	0.1337	0.1459	0.1937

Note. Robust t-statistics in parentheses. * *p* < 0.1; ** *p* < 0.05; *** *p* < 0.01.

## Data Availability

The data of this study is available from the corresponding author upon reasonable request.

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
