# Peer review of "Which Receives More Attention, Online Review Sentiment or Online Review Rating? Spillover Effect Analysis from JD.com"

_behavsci, 2024, doi:10.3390/bs14090823_

Round 1

Reviewer 1 Report

Comments and Suggestions for Authors

Dear Authors,

Please find my recommendations for improving your paper.

Introduction and Literature Review - Ensure these sections are concise and directly address the research questions. Incorporate more up-to-date resources not only in these sections but also in the discussion part of your paper.

Data and Methodology - Conduct additional robustness checks to validate your findings. Also, focusing exclusively on one, particularly specific, category of goods is a highly limiting factor.

Discussion  - Clearly articulate the theoretical contributions of your study within the context of the existing literature. Most  importantly, compare your results with those of other (recent) studies to highlight the uniqueness and significance of your findings.

I hope these recommendations will help you in refining your paper.

Unfortunately at this stage I cannot recommend your article for publication.

Reviewer 2 Report

Comments and Suggestions for Authors

Please provide the complete results of the potential Dirichlet allocation calculations, including the basis for determining the number of topics. Did you calculate the coherence and perplexity values? Is there any overlap between the topics? What are the probability values, counts, and weights of the vocabulary within each topic? Do any words contribute to multiple topics?

Please provide the complete results of the sentiment analysis and upload them as an attachment. As far as I know, SnowNLP has limited capability in recognizing sentiment in Chinese. How did the authors address the issue of discrepancies between sentence meaning and sentiment scores? For example, how did they handle cases where the overall context is positive, but the use of some words classified as negative results in a very low overall sentiment score?

Table 1 lists "quality" twice. What do title1, title2, and title4 refer to?

In the regression analysis, there are multiple coefficients below 0.1, with some as low as around 0.02. Is it reasonable for the authors to still consider these as significant results?

Should the distribution of variables be checked for assumptions before conducting the analysis?

Should the mentioned control variables be moderator variables instead?

Results and discussion should be separated.

Additionally, I have a concern: the corresponding author’s profile on ResearchGate lists Diagnostic Imaging, and her research focus before joining Beihang University was related to medicine. Notably, the article “A Multiorgan Segmentation Model for CT Volumes via Full Convolution-Deconvolution Network,” published in Biomed Research International before 2024 by Hindawi, is particularly worth noting.

Round 2

Reviewer 1 Report

Comments and Suggestions for Authors

I thank the authors for incorporating my comments. The article, in its current form better communicates the findings of the research conducted. At the same time, the added value to the scientific community has increased. 

After carefully reading the article, I have no further comments on the paper's editing.

I recommend the paper for publication.

Author Response

We are grateful to the anonymous reviewer for
critically reading this article and for giving important suggestions to improve this article.

Reviewer 2 Report

Comments and Suggestions for Authors

My primary concern is with the statistical results presented in the manuscript. Specifically, the author has reported several regression coefficients smaller than 0.1 that are highly significant, which is quite unusual in regression analyses. Such small coefficients, especially when coupled with high significance, raise questions about the robustness of the results or the appropriateness of the model used.
